# Mesothelin Expression Is Not Associated with the Presence of Cancer Stem Cell Markers SOX2 and ALDH1 in Ovarian Cancer

**DOI:** 10.3390/ijms23031016

**Published:** 2022-01-18

**Authors:** Mariana Nunes, Francisca Pacheco, Ricardo Coelho, Dina Leitão, Sara Ricardo, Leonor David

**Affiliations:** 1Differentiation and Cancer Group, Institute for Research and Innovation in Health (i3S) of the University of Porto/Institute of Molecular Pathology and Immunology of the University of Porto (Ipatimup), 4200-135 Porto, Portugal; mnunes@ipatimup.pt (M.N.); franciscapacheco19@icloud.com (F.P.); sricardo@ipatimup.pt (S.R.); 2Institute of Biomedical Sciences Abel Salazar (ICBAS), University of Porto, 4050-313 Porto, Portugal; 3Department of Pathology, Medical Faculty of the University of Porto (FMUP), 4200-319 Porto, Portugal; dinaraquel@med.up.pt; 4Ovarian Cancer Research, Department of Biomedicine, University Hospital Basel and University of Basel, CH-4031 Basel, Switzerland; ricardojorge.bouca-novacoelho@unibas.ch; 5TOXRUN, Toxicology Research Unit, University Institute of Health Sciences, CESPU, CRL, 4585-116 Gandra, Portugal

**Keywords:** high-grade serous carcinoma, mesothelin, SOX2, ALDH1, cancer stem cells

## Abstract

Mesothelin (MSLN) overexpression (OE) is a frequent finding in ovarian carcinomas and increases cell survival and tumor aggressiveness. Since cancer stem cells (CSCs) contribute to pathogenesis, chemoresistance and malignant behavior in ovarian cancer (OC), we hypothesized that MSLN expression could be creating a favorable environment that nurtures CSCs. In this study, we analyzed the expression of MSLN and CSC markers SOX2 and ALDH1 by immunohistochemistry (IHC) in different model systems: primary high-grade serous carcinomas (HGSCs) and OC cell lines, including cell lines that were genetically engineered for MSLN expression by either CRISPR-Cas9-mediated knockout (Δ) or lentivirus-mediated OE. Cell lines, wild type and genetically engineered, were evaluated in 2D and 3D culture conditions and xenografted in nude mice. We observed that MSLN was widely expressed in HGSC, and restricted expression was observed in OC cell lines. In contrast, SOX2 and ALDH1 expression was limited in all tissue and cell models. Most importantly, the expression of CSC markers was independent of MSLN expression, and manipulation of MSLN expression did not affect CSC markers. In conclusion, MSLN expression is not involved in driving the CSC phenotype.

## 1. Introduction

Mesothelin (MSLN) is a glycosylphosphatidylinositol-linked protein [1] that is overexpressed in different tumor types, including ovarian cancer (OC) and mesotheliomas, and variably impacts the patient’s prognosis [2]. In a previous study, we observed that MSLN was overexpressed in OC, and overexpression (OE) was more significant in high-grade serous carcinoma (HGSC) [3]. In addition, MSLN OE was associated with the presence of ascites and poor progression-free survival [3]. These observations were supported by analysis of the Cancer Genome Atlas database, three independent transcriptomic datasets and protein expression data evaluated by immunohistochemistry (IHC) [3].

Cancer stem cells (CSCs) have been implicated in different steps of carcinogenesis, chemoresistance and cancer dissemination [4,5]. In HGSC, the population with cancer stem-like features represents a small subpopulation of undifferentiated cells with unique abilities such as self-renewal, proliferation, differentiation and multipotency that can partly explain the aggressive nature of OC [6], namely, dissemination in the peritoneal cavity, chemoresistance and recurrent disease [7,8]. However, the literature on CSC markers is full of inconsistencies due to both different methodologic approaches and the definition of CSC populations. These difficulties place limitations on interpreting results and limit their relevance and accuracy, as recognized in the review paper by Roy and Cowden Dahl [9]. Our decision to select SOX2 and ALDH1 was based on critical observations and on their reliable detection by IHC with existing antibodies. Some markers for CSCs have been identified and, among many others, include pluripotency-associated transcription factors, such as sex-determining region Y-box (SOX2) and detoxifying enzymes, e.g., aldehyde dehydrogenase 1 (ALDH1) [10,11]. 

SOX2 is a transcription factor that regulates embryonic development and stem cell maintenance [12,13]. Considering its importance in stem cells, SOX2 has also been studied in cancer to determine its potential role in tumor initiation, maintenance and targeted therapy [14,15]. SOX2 expression can be detected in 10–60.5% of ovarian carcinomas and has been associated with poor clinical outcome [12,16,17]. ALDH1 is involved in detoxification and the elimination of oxidative stress [18] through the oxidation of aldehydes into carboxylic acids [11]. Moreover, cells with high ALDH1 activity showed increased potential for self-renewal and stress resistance [7,19]. SOX2 and ALDH1 expression have been used as stemness markers for detecting cancer cell proliferation, migration, invasion and metastasis [20,21]. SOX2 and ALDH1 OE was found in cancer cells, particularly in HGSC, with higher tumorigenicity and invasion abilities [22,23]. In addition, their expression was correlated with chemoresistance and poor prognosis [23,24]. 

We hypothesized that the association of high MSLN expression with increased cell survival and peritoneal dissemination of OC cells [3] might be dependent on a surge in a population of CSCs nurtured by a microenvironment in which MSLN-positive cells could have a role. We centered our study on two accepted CSC markers, SOX2 and ALDH1, since we have extensive experience on their expression profiles with antibodies that we are familiar with. To test our assumption, we analyzed the *in situ* expression patterns of MSLN and CSC markers SOX2 and ALDH1 in HGSC tissues and OC cell lines. We also evaluated the expression of those markers in OC cell lines and their corresponding nude mice xenografts manipulated to express MSLN, either by CRISPR-*Cas9*-mediated homozygous Δ or lentivirus-mediated OE. Double IHC with MSLN and CSC markers was performed to potentially identify specific subsets of CSC-like populations associated with MSLN expression. 

## 2. Results

### 2.1. MSLN Is Not Associated with Expression of CSC Markers in HGSC

Despite the high expression of MSLN, we frequently observed focal areas of negative staining in HGSC cases. This represented an ideal setting to see how CSC markers behaved in MSLN-positive and -negative cells. We selected whole sections from 14 HGSCs from a previously published series [25] to evaluate the positivity of MSLN versus CSC markers by IHC. In these 14 cases, we studied all tissue areas in order to better identify tissue heterogeneity in the expression of these biomarkers. MSLN was expressed at high levels (76–100%) in 11/14 cases, and in 3/14 cases, 100% of tumor cells presented MSLN positivity (Figure 1A). In most cases (n = 11), focal areas of MSLN negativity were identified (Figure 1A–C). 

Then, serial sections were evaluated for coexpression of MSLN and CSC markers SOX2 and ALDH1 in the 14 HGSC cases. No clear association was observed in the overall series, as expected, since all cases were positive (76–100%) for MSLN (Figure 2A). In most cases, SOX2 was negative (11/14 cases) or very focally expressed (about 20% of the cancer cells in 3/14 cases), and ALDH1 was, in most cases (11/14 cases), also expressed in at least 20% of the cancer cells (Figure 2A) (see color key). Not surprisingly, most positive areas for SOX2 or ALDH1 were colocalized with MSLN-expressing cells (Figure 2B), and, accordingly, no significant association was identified between the expression of MSLN, SOX2 and ALDH1.

The HGSC case shown in Figure 3A displays an area that is positive for MSLN and ALDH1 and mostly negative for SOX2, whereas a focal area (square) of positive cells for SOX2 is also positive for MSLN and ALDH1. In Figure 3B, a focal area (square) positive for SOX2 and MSLN is negative for ALDH1. 

Finally, Figure 4 highlights the random correlation between the expression of the three markers, with SOX2 and ALDH1 only focally coinciding and MSLN similarly colocalizing with the CSC markers, but mostly extending beyond areas stained for SOX2 and/or ALDH1.

### 2.2. MSLN Is Not Associated with Expression of CSC Markers in OC Cell Lines

Most of the OC cell lines used in preclinical studies fail to represent the features of HGSC, namely, by showing limited expression of MUC16 and truncated O-glycans [25]. Through the evaluation of seven OC cell lines for the current markers, we observed that only two (OVCAR3 and OVCAR8) were highly positive for MSLN, again showing that cell lines reproduce the defective pattern of expression observed in HGSC in both 2D and 3D culture conditions (Figure 5A). Curiously, SOX2 was highly expressed in a single cell line, OVCAR3, that also expressed high amounts of MSLN but rarely expressed ALDH1 (Figure 5B). On the other hand, ALDH1 was highly expressed in an OC cell line, IGROV1, that was negative for MSLN and SOX2 expression (Figure 5C).

### 2.3. Genetic Engineering of MSLN Does Not Affect CSC Marker Expression

OVCAR3 and OVCAR 8 Δ*MSLN* did not show any meaningful difference in CSC marker expression as compared with the corresponding parental cells in both 2D cell cultures (Figure 6A) and nude mice xenografts (Figure 6B) (both 6A and 6B show only OVCAR8). The same applies to BG1 MSLN OE and OVCAR4 MSLN OE cells as compared to parental cells in 2D cell cultures (Figure 7A). BG1 MSLN OE cells had an increase in SOX2 expression when implanted in nude mice, independent of MSLN OE, and conversely, decreased expression of SOX2 was observed in nude mice with tumors derived from OVCAR4 MSLN OE (Figure 7B) (both 7A and 7B show only BG1). Appendix A shows additional data using CSC markers CV44v6 and OCT4A. Similar to what was observed with SOX2 and ALDH1, no meaningful differences were observed between parental and Δ or OE cells (Appendix A). Increased expression of CD44v6 was observed in OVCAR3 Δ*MSLN* cells (Appendix A). No other differences were observed for either CD44v6 or OCT4 in BG1 as compared to BG1 MSLN OE cells (Appendix A).

## 3. Discussion

Previous studies from our and other groups indicate that MSLN OE is associated with improved cell survival and invasiveness [3,26,27], increased tumor burden and peritoneal dissemination [3,28], as well as poor prognosis in OC patients [28,29,30]. Some authors have linked the OE of MSLN to increased expression of CSCs [31,32], which could thus contribute to explaining the association of MSLN with cell aggressiveness and cancer progression.

The role of CSC markers in cancer behavior is well established [14]; however, in OC, their functionality is still disputable [33]. Similarly, the role of CSCs in OC chemoresistance is an issue open to discussion [34], but studies support their use as potential therapeutic targets [35]. We took into consideration the results from He and Matsuzawa [31,32] and our own observations implicating MSLN in the aggressive behavior of OC cells and endeavored to study, in different models, the relationship between the expression of MSLN and CSC markers SOX2 and ALDH1. 

Our results indicate that MSLN expression is not associated, in HGSC, with the presence or absence of CSC markers SOX2 or ALDH1. Similar to findings in human carcinomas, expression of MSLN was not associated with the presence or absence of CSC markers in OC cell lines. These results were reinforced by the study of OC cell lines engineered for MSLN expression. In both MSLN Δ and overexpressing OC cells, no association was found between the expression of MSLN and SOX2 and/or ALDH1 CSC markers, nor was it associated with OCT4A and CD44v6 (Appendix A). We tested CD44v6 and OCT4A as additional CSC markers since they are also shown to be relevant in the OC setting. CD44v6 is widely accepted as a CSC marker in solid tumors, and in OC, it was also demonstrated to be present in a metastasis-initiating population [36]. OCT4A affects CSC sphere formation of OC cell lines [37]. The results that we obtained with CD44v6 and OCT4A did not contradict but also did not help to clarify our data for SOX2 and ALDH1. Our SOX2 and ALDH1 results contrast with those obtained using Δ and OE MSLN models from lung cancer cell lines [31]. These models agree with our own observations on the role of MSLN in cell migration and invasion as well as tumorigenicity [3]. However, the study by He et al. showed that MSLN was associated with the CSC phenotype, as evaluated by ALDH1 expression [31], which is not supported by the current work. Matsuzawa and collaborators also showed that MSLN blockade by Amatuximab increases c-met expression, another CSC marker, in pancreatic cancer cell lines, suggesting an indirect link between MSLN and the CSC phenotype [32]. Overall, our study suggests that the increased cell survival and aggressive behavior associated with MSLN expression are not dependent on the hypothesized role of MSLN in nurturing the CSC phenotype.

Our double-staining results in HGSC cases revealed not only co-occurrence but also coexpression of SOX2 and ALDH1 in localized areas and cells. These observations somewhat fit with what has been described as the coexistence of cell clusters with different proliferation capacities and hence different microenvironmentally modulated phenotypes [38]. Both SOX2 and ALDH1 were coexpressed with MSLN in focal areas, but most MSLN-positive cells were negative for both markers, suggesting that MSLN is not associated with the expression of CSC biomarkers. Only a minority of cases showed co-occurrence of all three markers, and very few cases showed their coexpression in the same cell populations. We can conclude that MSLN expression is not topographically associated with SOX2 and ALDH1 expression at the cellular level. 

The selection of CSC markers is a nontrivial subject since many doubts exist as to their relative relevance and accuracy. Our decision to select SOX2 and ALDH1 was based on critical observations and on their reliable detection by IHC with existing antibodies. SOX2 expression was confirmed as a CSC marker by several authors based on its capacity to induce CSC properties, including stemness, as evidenced by tumor-initiating capacity, sphere formation, selective chemoresistance and promotion of in vivo tumorigenicity [12,13,39,40]. In addition, SOX2 OE potentiates tumor aggressiveness through the enhanced migration and invasion capacity of OC cell lines [41,42] and increases tumor size in in vivo models [12]. Selection of ALDH1, a quasi-universally accepted biomarker, was particularly attractive due to its very interesting role as a driver for chemoresistance and therapeutic targeting [43]. This last aspect, suggesting that ALDH1 is a CSC marker strongly associated with chemoresistance and an adverse clinical course, is clearly at stake in OC [8,44,45,46]. 

Interestingly, Fisher et al. demonstrated the increased expression of CSC markers SOX2 and ALDH1 in HGSC [11]. Moreover, despite the common co-occurrence of these markers in the OC setting, coexpression is rare and limited in HGSC [11], which is in agreement with our current observations.

## 4. Materials and Methods

### 4.1. Patient Samples

A retrospective series of 20 HGSC patients, diagnosed between 2002 and 2015, was retrieved from the archives of the Pathology Department of Centro Hospitalar de São João (CHSJ), Porto, Portugal, and analyzed in a previous study [3]. A subset of 14 HGSC samples was selected to study whole-tumor sections. This series was selected based on the quality and representability of the histological material, clinical information and histological type (WHO classification). All human samples were selected in accordance with local ethical guidelines (as stipulated by the Declaration of Helsinki) and approved by the Ethical Committee from CHSJ (Ref.86/2017).

### 4.2. Cell Lines

MSLN-manipulated cell lines were previously generated in our laboratory [3]. Briefly, two MSLN^high^ (OVCAR3 and OVCAR8) and two MSLN^low^ (OVCAR4 and BG1) OC cell lines were selected in order to generate Δ*MSLN* and cells with constitutive MSLN OE, respectively. Δ*MSLN* was obtained by CRISPR-*Cas9* gene-editing technology, and MSLN OE was induced by lentiviral transduction through pUltra-MSLN for biscistronic expression of MSLN and EGFP proteins [3]. 

OC cell lines were cultured under 2D and 3D conditions in RPMI 1640 (Thermo Fisher Scientific, Waltham, MA, USA) containing 10% fetal bovine serum (FBS) (Biowest, Nuaillé, France) and maintained at 37 °C and 5% CO_2_. For 3D cultures, 96-well round-bottom plates were coated with polyHEMA (poly(2-hydroxyethyl methacrylate)) (Sigma-Aldrich, St. Louis, MO, USA) at 120 mg/mL in 95% ethanol. Aggregates were generated by plating 4 × 10^3^ cells/well and incubating them at 37 °C and 5% CO_2_ for 96 h. 

All cell lines were authenticated using short tandem repeat (STR) profiling (PowerPlex 16 HS kit, Promega, Madison, WI, USA) and regularly tested for the absence of mycoplasma. IGROV1 was included in our panel of OC cell lines, originally described as representing HGSC but recently suggested as better representing an endometrioid carcinoma with a hypermutated profile and co-occurrence of PIK3CA and PTEN mutations [47].

### 4.3. Nude Mice Xenografts 

The xenograft models were previously generated in our laboratory by intraperitoneal injection in nude mice (N:NIH(S) II: *nu*/*nu* mice) using several OC cell lines manipulated to express MSLN (i.e., cell lines described in Section 4.2) [3]. Hematoxylin and eosin (H&E) staining from Douglas pouch and peritoneum implants were examined under an optical microscope to evaluate tumor histologic characteristics. All immunostaining procedures in nude mice tissues were performed according to the protocol described in Section 4.5.

### 4.4. Cell/Tissue Microarray Construction 

All cell lines were cultured in 2D and 3D conditions and arrayed in a cell microarray (CMA) block designed and constructed as previously described [48,49,50]. Briefly, 2D cultures were collected by scraping cells from culture dishes in PBS 1x, and 3D cultures were simply aspirated from each well, followed by centrifugation and fixation with 10% (*v*/*v*) neutral buffered formalin (AppliChem, Darmstadt, Germany). After fixation, cells were resuspended in HistoGel^TM^ (Thermo Fisher Scientific, Waltham, MA, USA) according to the manufacturer’s instructions, followed by standard histological processing and paraffin embedding. Each cell line block (donor block) was sectioned and stained with H&E for morphology control.

The CMA for 2D and 3D cell culture conditions and the tissue microarray (TMA) for nude mice xenografts were designed and constructed by adding one core (1.5 mm in diameter) from each donor block to a recipient paraffin block. Tumor tissue cores were included as controls. After construction, CMA was homogenized at 37 °C overnight and sectioned with a standard microtome at 3–4 µm thickness.

### 4.5. Immunohistochemistry 

IHC for HGSC tissues and OC cell lines was performed using a manual system. After deparaffinization and hydration, heat-induced (98 °C) antigen retrieval was performed with citrate buffer solution (1:100 at pH 6.0; ThermoFisher Scientific, Waltham, MA, USA) or ethylenediaminetetraacetic acid (EDTA; 1:100; ThermoFisher Scientific, Waltham, MA, USA). The detection system used was Dako REAL^TM^ EnVision^TM^ Detection System Peroxidase/DAB+, Rabbit/Mouse (Agilent Dako, Santa Clara, CA, USA) kit, which was used according to the manufacturer’s instructions. Briefly, endogenous peroxidase activity was blocked with 3% (*v*/*v*) hydrogen peroxide solution (ThermoFisher Scientific, Waltham, MA, USA). Slides were immunostained with monoclonal antibodies for MSLN (1:50, SP74, ThermoFisher Scientific, Waltham, MA, USA), SOX2 (1:25, SP76, Cell Marque, California, CA, USA) and ALDH1 (1:200, D9Q8E, Cell Signaling Technology, Massachusetts, MA, USA) and incubated for 1 h at room temperature (RT). Additional antibodies, CD44v6 (1:100; 336700, Life Technologies, Carlsbad, CA, USA) and OCT4A (1:100, C52G3, Cell Signaling Technology, Danvers, MA, USA), were incubated for 1 h at RT as shown in Appendix A. Primary antibodies were detected using a secondary antibody with horseradish peroxidase (HRP)-labeled polymer, and visualization of the reaction was performed using diaminobenzidine according to the manufacturer’s instructions. 

Double IHC for HGSC tissues was performed using the EnVision^TM^ G|2 Doublestain System, DAB+/Permanent Red Rabbit/Mouse (Agilent Dako, Santa Clara, CA, USA) kit according to the manufacturer’s instructions. Briefly, endogenous peroxidase activity was blocked, slides were immunostained with the first monoclonal antibody, i.e., SOX2 (1:25, SP76, Cell Marque, Rocklin, CA, USA) and incubated for 1 h at RT. This primary antibody was detected using a secondary antibody with HRP-labeled polymer, and visualization of the reaction was performed using diaminobenzidine according to the manufacturer’s instructions. Then, slides were immunostained with the second monoclonal antibody, i.e., MSLN (1:50, SP74, ThermoFisher Scientific, Waltham, MA, USA) or ALDH1 (1:200, D9Q8E, Cell Signaling Technology, Danvers, MA, USA), and incubated for 1 h at RT. These second antibodies were detected using a secondary antibody with alkaline phosphatase–labeled polymer, and visualization of the reaction was performed using permanent red solution according to the manufacturer’s instructions. 

IHC for TMAs of nude mice tissues was performed using an automated Ventana BenchMark ULTRAStaining System using the OptiView DAB IHC Detection Kit (Roche/Ventana Medical Systems, Oro Valley, AZ, USA) according to the manufacturer’s instructions. The dilutions of primary antibodies were the same as mentioned above and were added manually with the Ventana™ BenchMark ULTRA platform, according to the manufacturer’s instructions. These second antibodies were detected using the OptiView™ Universal DAB detection kit (Ventana Medical Systems, Oro Valley AZ, USA). A negative control slide was used in place of the primary antibody to evaluate nonspecific staining using a specific reagent, i.e., rabbit monoclonal negative control Ig (Ventana Medical Systems, Oro Valley, AZ, USA). 

Nuclear staining with hematoxylin was performed, and slides were dehydrated, clarified, and sealed with coverslips using a permanent mounting medium for optical microscope analysis.

IHC results were evaluated by four independent observers (L.D., S.R., F.P. and M.N.), who registered the percentage of cells stained (0–10% (considered negative), 11–25%, 26–50%, 51–75% and 76–100%). 

## 5. Conclusions

We observed that 79% of the HGSC cases had an MSLN-negative subpopulation, and this observation could be relevant at a moment when MSLN-targeted therapies are being implemented. CSC markers SOX2 and/or ALDH1 were focally identified in all HGSC cases in a manner unrelated to MSLN expression. Similarly, MSLN expression was not associated with either SOX2 or ALDH1 expression in cell lines, and MSLN OE or Δ in genetically engineered OC cell lines did not influence SOX2 and/or ALDH1 expression. In conclusion, MSLN expression is not involved in driving a microenvironment that nurtures CSCs.

## Figures and Tables

**Figure 1 ijms-23-01016-f001:**
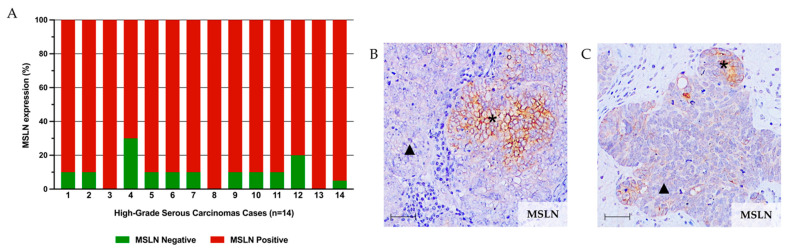
Mesothelin (MSLN) expression pattern in high-grade serous carcinoma (HGSC) series. (**A**) Bar chart showing the percentage of positive (red) and negative (green) cells for MSLN expression in 14 HGSC cases. (**B**,**C**) Representative immunohistochemistry (IHC) images for MSLN-positive (asterisk) and MSLN-negative (triangle) areas. Scale bar: 100 μm.

**Figure 2 ijms-23-01016-f002:**
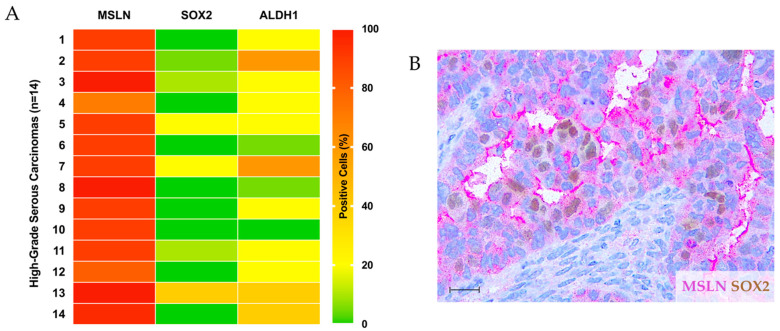
MSLN and cancer stem cell (CSC) markers SOX2 and ALDH1 expression in HGSC series. (**A**) Heat map showing the percentage of positive cells for MSLN and cancer CSC markers SOX2 and ALDH1 in 14 HGSC cases. Color key represents the percentage of positive cells for each marker. (**B**) Representative double IHC staining image for MSLN (pink) and SOX2 (brown). Scale bar: 100 μm.

**Figure 3 ijms-23-01016-f003:**
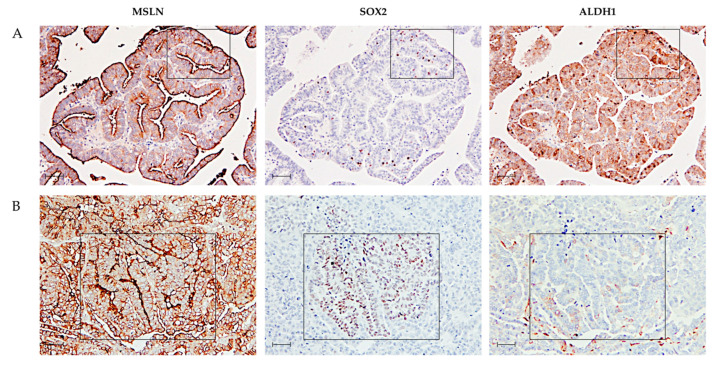
Representative IHC images for MSLN, SOX2 and ALDH1 expression patterns in two HGSC cases. (**A**) In this case, the majority of tumor cells are positive for MSLN and ALDH1 and negative for SOX2. A focal area (square) of SOX2-positive cells is also positive for both MSLN and ALDH1. (**B**) In this second example, most tumor cells are positive for MSLN and negative for ALDH1, and a focal area is positive for SOX2. The selected area (square) is positive for MSLN and SOX2 and negative for ALDH1. Scale bar: 100 μm.

**Figure 4 ijms-23-01016-f004:**
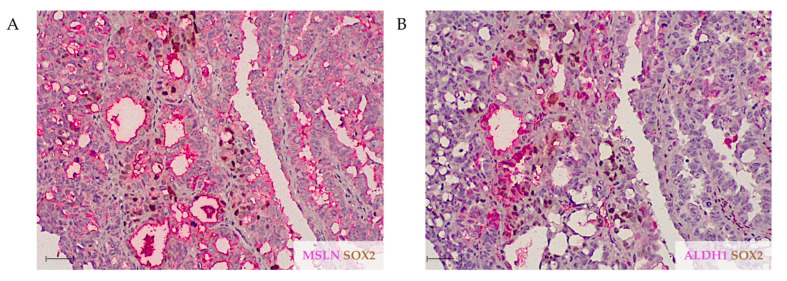
Representative double IHC images for MSLN/SOX2 and ALDH1/SOX2 expression in a selected HGSC case. (**A**) The majority of tumor cells are positive for MSLN (pink) with focal areas of SOX2 positivity (brown). (**B**) ALDH1-positive (pink) area coincides with SOX2-positive (brown) cells. Scale bar: 100 μm.

**Figure 5 ijms-23-01016-f005:**
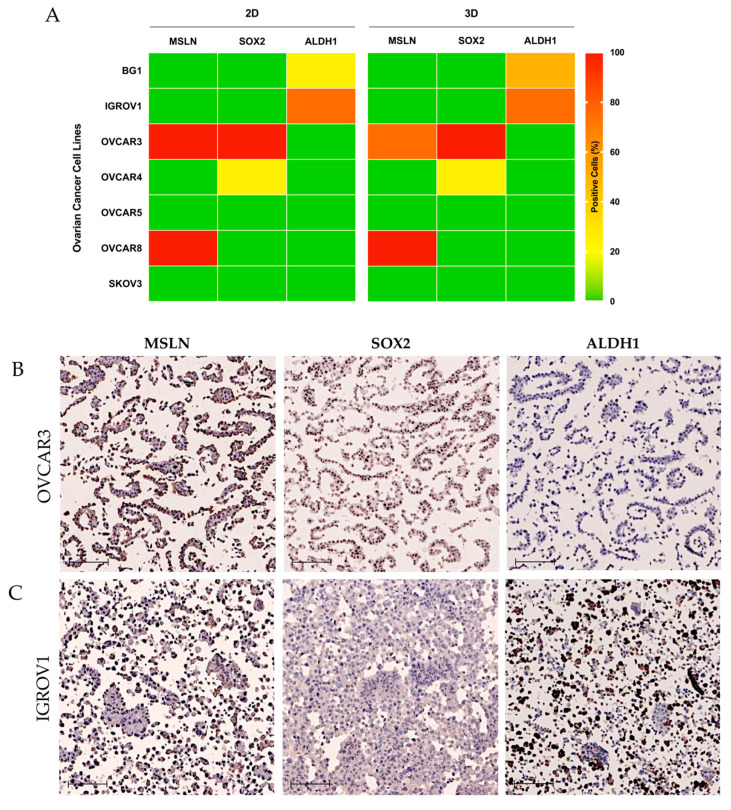
MSLN and CSC markers SOX2 and ALDH1 expression in ovarian cancer (OC) cell lines. (**A**) Heat map showing the expression of MSLN, SOX2 and ALDH1 in a panel of OC cell lines. Expression profile obtained by IHC in 7 OC cell lines cultured under 2D and 3D conditions. Color key represents the percentage of positive cells for each marker. (**B**,**C**) Representative IHC images for MSLN, SOX2 and ALDH1 expression patterns in OVCAR3 (**B**) and IGROV1 (**C**) cell lines in 2D culture conditions. Scale bar: 50 μm.

**Figure 6 ijms-23-01016-f006:**
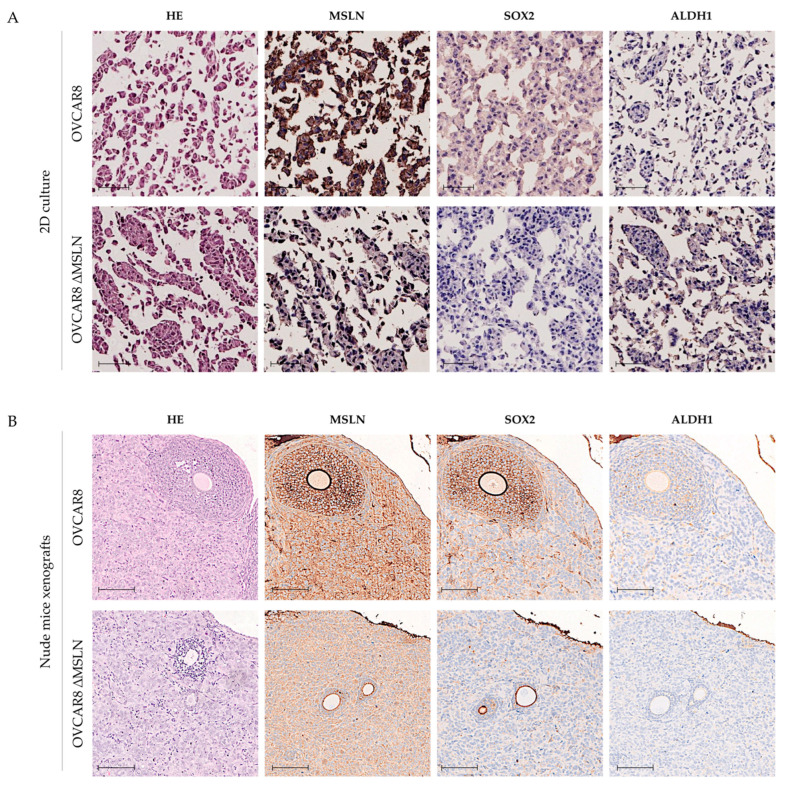
MSLN and CSC markers SOX2 and ALDH1 expression in OVCAR8 cell line with knockout (Δ)*MSLN*. Representative hematoxylin and eosin (H&E) and IHC images for MSLN, SOX2 and ALDH1 expression in OVCAR8 and OVCAR8 Δ*MSLN* cells in 2D culture conditions (**A**) and in nude mice xenografts (**B**). Scale bars: 50 μm (**A**) and 100 μm (**B**).

**Figure 7 ijms-23-01016-f007:**
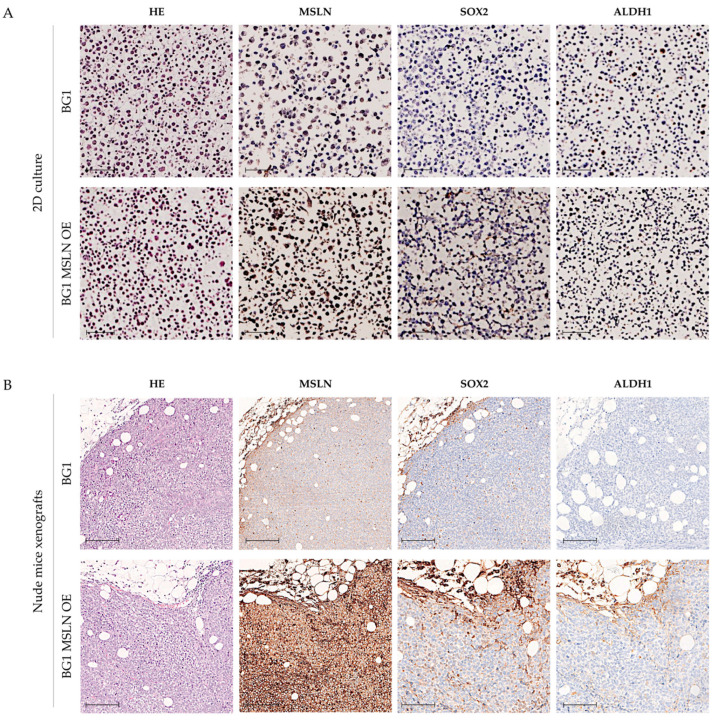
MSLN and CSC markers SOX2 and ALDH1 expression in BG1 cell line with MSLN OE. Representative H&E and IHC images for MSLN, SOX2 and ALDH1 expression in BG1 and BG1 MSLN OE cells in 2D culture conditions (**A**) and in nude mice xenografts (**B**). Scale bars: 50 μm (**A**) and 100 μm (**B**).

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
