# Peer review of "Mesothelin Expression Is Not Associated with the Presence of Cancer Stem Cell Markers SOX2 and ALDH1 in Ovarian Cancer"

_ijms, 2022, doi:10.3390/ijms23031016_

Round 1

Reviewer 1 Report

This is an interesting study concerning mesothelin expression in ovarian cancer (OC) tissues and cell lines, analyzed in relation to SOX2 and ALDH1 co-occurrence and co-expression. In addition, the influence of mesothelin overexpression and down-regulation, by genetic engineering, on the expression of SOX2 and ALDH1 was analyzed both in vitro and in vivo.

However, in my opinion conclusions are too far reaching and the title is not adequate.

I would insist on changing the title, e.g. for the following: Mesothelin expression is not associated with the presence of cancer stem cell markers SOX2 and ALDH1 in ovarian cancer.

First of all, SOX2 and ALDH1 are only two from many (a dozen or so) proposed stem cell markers in OC. Supposition that SOX2+/ALDH1+ OC cells have stem-like potential is unproven.  NB. the literature on the topic of OC CSC much wider than Authors cite!

Secondly, cancer stem cell theory postulates that tumor is comprised of (heterogeneous) cancer cell populations and a very tiny fraction of CSC population(s). I have the impression that Authors expect that CSC population(s) is very abundant – they observe mesothelin expression in 80-100% of cells in OC sections and propose that these cells have stem-like properties. This is wrong.

In fact, Authors only signalize, and they do it very late, at the end of Discussion, how complicated the problem of CSC definition and detection of CSC in OC is. I would like this problem to be signalized in the Introduction. I would also ask Authors to give an explanation concerning their choice of (potential) CSC markers (Sox2, ALDH1) they focus on.

In fact, in ovarian cancer there is no consensus concerning stem-like cell markers.  In the review by (Roy and Cowden Dahl, 2018) at least 13 putative markers for ovarian cancer stem-like cells are discussed, eg: CD44, CD133, CD117, CD73, CD24, CD90, EpCAM, ALDH1. Roy, L.; Cowden Dahl, K. D. Can stemness and chemoresistance be therapeutically targeted via signaling pathways in ovarian cancer? Cancers 2018, 10(0241), 1-23. DOI: 10.3390/cancers10080241 

In the literature there are many phenotypes proposed for ovarian cancer stem-like cells, e.g. CD133+/CD117+ or CD44+/CD133+/CD24-, but also CD44+/CD133+/CD24+.

Due to ambiguous results of the studies concerning identification of these markers, it is postulated that patients may have more than one pool of stem-like cells and/or different patients may have stem-like cells with distinct phenotypes.

In particular, Parte et al described the existence, distribution and abundance of various stem cell populations in normal ovary and CSCs in ovarian tumors at various stages of tumorigenesis using several relevant markers, Parte, S.C., Batra, S.K. & Kakar, S.S. Characterization of stem cell and cancer stem cell populations in ovary and ovarian tumors. J Ovarian Res 11, 69 (2018). https://doi.org/10.1186/s13048-018-0439-3

Additionally, I would like to alert Authors that IGROV1 cell line is not a model of HGSOC. It has been shown by Domcke et al by genetic analysis that IGROV1  is most probably of endometrioid or clear cell rather than high-grade serous origin.

Domcke, S., Sinha, R., Levine, D. et al. Evaluating cell lines as tumour models by comparison of genomic profiles. Nat Commun 4, 2126 (2013). https://doi.org/10.1038/ncomms3126

As to the second cell line used BG1, it is described in Cellosaurus as a problematic cell line: Partially contaminated. Some stocks are contaminated by MCF-7 (PubMed=22710073; PubMed=25321415). It is recommended to run an STR similarity search on this cell line https://web.expasy.org/cellosaurus/CVCL_6570 

To sum up - I would prefer the Title to be changed and the Introduction to be rewritten according to my remarks

Reviewer 2 Report

In this manuscript the authors studied the role of MSLN overexpression in driving of CSC phenotype in ovarian carcinomae. Although the results contain useful information, the manuscript as a whole is descriptive and lacks inspirable messages for the readers. The reviewer cannot recommend this manuscript as a candidate for publication in IJMS journal in the current version. The reviewer suggests that authors should perform additional experiments to show the significance of the results. For example, the expression of CSC markers in cell culture experiments should be confirmed by immunoblots with appropriate antibodies. The choice of OC CSC markers seems to be ambiguous. While SOX2 transcription factor was shown to be a driver of recurrence which could serve as a reliable marker of OC CSC with high relapse potential, the expression of ALDH1 depends on exact OC cell line, so it is not entirely correct to consider it as OC stem cell marker.

Round 2

Reviewer 1 Report

I have recieved amended version of the manuscript by Mariana Nunes et al. I appreciate the effort Authors have made to fulfill my requirements. I am satisfied with the changes made in the manuscript and with explanations given in the coverletter. I think that the manuscript has gained a lot. In particular, I appreciate adding new results, concerning CD44v6 and OCT4A, to the supplement. (NB. it is interesting that downregulation of MSLN in OVCAR3 caused very intense membrane expression of CD44v6; CD44 is frequently regared as a negative prognostic marker in OC). I also like the fact that Authors based their study on the use of validated antibodies with which they are familiar. To sum up, in my opinion the manuscript is now acceptable for publishing in IJMS.